# Neural network-based approach to phase space integration

**Matthev D. Klimek[1,2⋆] and Maxim Perelstein[1]**

**1** Laboratory for Elementary Particle Physics, Cornell University, Ithaca, NY, USA
**2** Department of Physics, Korea University, Seoul, Republic of Korea

⋆ klimek@cornell.edu

## Abstract

Monte Carlo methods are widely used in particle physics to integrate and sample probability distributions on phase space. We present an Artificial Neural Network (ANN) algorithm optimized for this task, and apply it to several examples of relevance for particle physics, including situations with non-trivial features such as sharp resonances and soft/collinear enhancements. Excellent performance has been demonstrated, with the trained ANN achieving unweighting efficiencies between 30% – 75%. In contrast to traditional algorithms, the ANN-based approach does not require that the phase space coordinates be aligned with resonant or other features in the cross section.


# 1  Introduction

Monte Carlo (MC) techniques are widely used for sampling multi-dimensional probability distribution functions (pdfs) and for numerical integration in multi-dimensional spaces. The particular application studied in this paper occurs in elementary particle physics, where the outcome of a collision of two particles is described by a pdf, the differential cross section, which can typically be calculated within a Quantum Field Theory framework. Monte Carlo tools are used to generate a sample of outcomes ("events") realizing the relevant pdf. Comparing the MC samples with experimental data tests the underlying theory used to calculate the pdf. Monte Carlo techniques have been used in particle physics since at least the 1970's, and continue to play a crucial role in the analysis of data from colliders such as the Large Hadron Collider.

In this paper, we will explore the use of a Artificial Neural Network (ANN) as a MC generator. Previous work [1] demonstrated that ANN-based MC simulation can significantly outperform traditional algorithms, such as VEGAS [2,3], when applied to a toy problem of sampling multi-dimensional Gaussian pdfs. In this paper, we apply an ANN-based MC algorithm to examples of direct relevance for particle physics. Specifically, we will use ANN-based MC to simulate events for three-body decay of a generic scalar particle, including various resonance structures in the decay amplitude, and quark pair-production in electron-positron collisions, including the radiation of an extra gluon in the final state at parton level. These examples involve non-trivial issues relevant to phase space integration in particle physics, such as large enhancements of the pdfs in specific regions of phase space due to soft/collinear singularities and/or resonances. We also present a method for implementing kinematic cuts in our algorithm. Our work demonstrates that the ANN-based algorithm is not hobbled by these issues, opening the road to a broader application of such algorithms in particle physics contexts.

The basic idea of ANN-based MC simulation is sketched in Fig. 1. The ANN acts as a map between two $n$-dimensional hypercubes, the "input space" $\mathcal{I}$ and the "target space" $\mathcal{T}$. A set of points $\mathbf{x}_i \in \mathcal{I}$ is drawn randomly, with a flat distribution, from the input space. The idea is to construct an ANN such that these points are mapped onto a set $\mathbf{y}_i \in \mathcal{T}$ distributed according to a pre-specified pdf, $f(\mathbf{y})$, on the target space. The behavior of the ANN is controlled by a set of parameters $\mathbf{w}$. The extent to which the pdf induced on $\mathcal{T}$ by the ANN matches the desired pdf is quantified by a loss function $L(\mathbf{w})$. By taking the gradient of the loss function with respect to $\mathbf{w}$, one can infer what adjustments to the ANN's parameters would result in a better match with the desired pdf. The training of the ANN consists of iteratively adjusting the parameters and then calculating the loss function again with the new parameters until sufficiently good performance is achieved. This procedure will be described in more detail later in this paper.

Once the ANN is trained, it can be used to generate a raw Monte Carlo sample. This sample will resemble one distributed according to the target pdf to the extent that training has been able to minimize the loss function. The raw sample can then be refined by unweighting, that is, keeping only a fraction of the events in proportion to the ratio of the target pdf to the pdf induced by the ANN. The fraction of the total sample that is retained after this procedure is called the efficiency, and is a good measure of the performance of the ANN as a MC generator. This procedure is depicted in Fig. 2. The efficiency calculation is presented in more detail later in this paper.

In the application to particle collisions or decays, the target space is $N$-body phase space, where $N$ is the number of particles in the final state. The kinematics of the final state is described by the 3-momentum of each particle. These satisfy four constraints from energy-momentum conservation, so the dimensionality of this space is $n = 3N - 4$. The coordinates on phase space will be chosen so that it is a unit hypercube. The desired pdf $f(\mathbf{y})$ is the fully differential cross section in these coordinates, and the points $\mathbf{y}_i$ form the raw MC sample, to

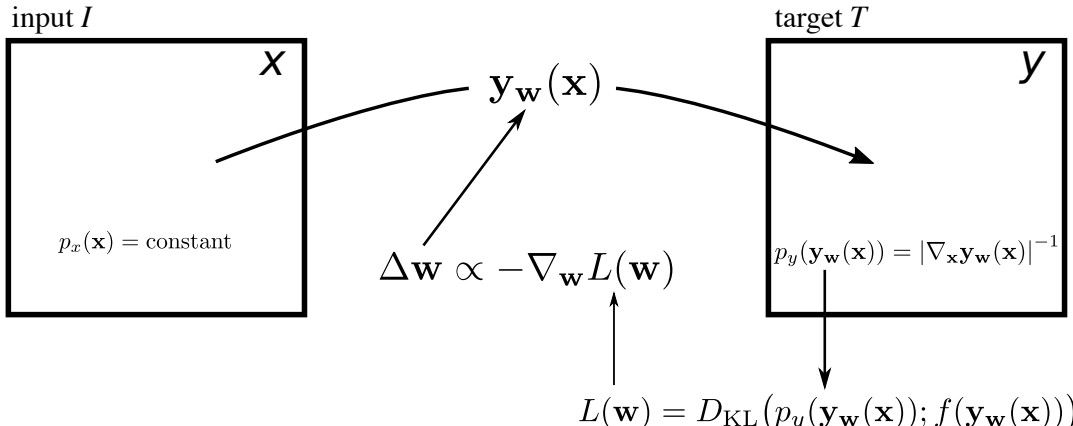

Figure 1: The ANN is a map $\mathbf{y_w(x)}$ between a uniformly sampled input space and a target space on which it induces a non-trivial pdf. During training, the parameters $\mathbf{w}$ of the ANN are adjusted by trying to decrease the value of a loss function $L(\mathbf{w})$ that quantifies the difference between the induced and the desired pdfs.

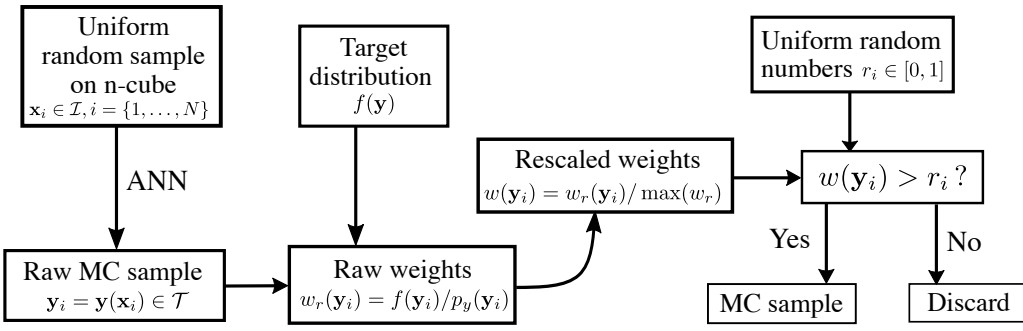

Figure 2: Diagram representing a Monte Carlo generator based on an ANN trained as described in the text. A raw sample is generated by the ANN and then refined by unweighting.

be distributed according to this cross section.

The VEGAS algorithm [2,3] is itself a form of machine learning. It approximates the desired pdf by dividing the domain of the pdf into discrete intervals and assigning each interval a constant value inversely proportional to its length, so that each contains the same amount of probability. Sampling from this approximation to the pdf then simply consists of randomly selecting an interval and sampling uniformly within that interval. The algorithm iteratively adjusts the edges of the intervals to better approximate the desired pdf.

This construction can also be viewed as map between an input space and a target space. Particularly, in this case it is a piecewise linear map, which can be seen as follows. Because all intervals contain the same probability, they have uniform size on the input space. However, each interval covers a different amount of the target space, in approximation to the non-trivial target pdf. In regions where the target pdf has large values, each interval covers only a small part of the target space, so the function that maps the input space onto the target space has

a small slope there. In regions where the target pdf is small and each interval covers a large portion, the slope is large. Moreover, because the sampling is uniform within each interval, the map must be piecewise linear. The ANN implementation discussed in the present work can simply be viewed as a continuum implementation of VEGAS. The ANN is a smooth, rather than piecewise linear, map between the input and target spaces, determined by the set of parameters **w**, which are adjusted to give a good approximation to the target pdf.

This provides two major advantages over traditional VEGAS. First, the use of a smooth map obviously allows for a much better approximation to the target pdf, which in turn gives better unweighting efficiency during event generation. Second, in multi-dimensional situations, VE-GAS uses a rectangular grid. Any sharp or rapidly varying features in the target pdf must be aligned with one of the grid axes in order for VEGAS to approximate it well. This requires one to inspect the target pdf first, and choose, if possible, coordinates that align with any sharp features that may be present. In cases with multiple sharp features, there may be no choice of coordinates that align with all of them, requiring the introduction of more sophisticated techniques. We will explore this issue in more detail in the main text, but we point out here that because the ANN is a smooth map, it has no intrinsic coordinate axes, and these issues are entirely avoided.

The rest of this paper is organized as follows. section 2 contains the details of the ANN and a complete description of the training procedure. In section 3, the ANN-based MC generator is applied to several sample problems of interest in particle physics. Conclusions are presented in Section 4. Finally, Appendix A contains details on the relative performance of various ANN architectures.

## 2 Setup

### 2.1 ANN Architecture and Features

We use a densely connected neural network (i.e., one in which each node is connected to every node in the preceding layer) with the same number of input and output nodes, equal to the phase space dimension. Various choices for the numbers of hidden layers and nodes per layer, collectively referred to as hyperparameters, have been tested. A comparison of the training performance under these choices is given in Appendix A. We find that larger ANNs train in fewer training epochs. A larger ANN takes longer to evaluate per data point. However, we hope this technique will be most useful in cases where the matrix element is very costly to evaluate. In these cases, the ANN evaluation time is subdominant, and a larger ANN that requires fewer matrix element evaluations to train is ideal.

We choose the simplest option of uniform sampling over a unit hypercube as the input. These inputs are to be mapped onto phase space, which we parametrize as a second hypercube in a coordinate system that will be described below.

The ANN is an event generator, that is, a map $\mathbf{y_w}(\mathbf{x})$ between points in an input space and points in phase space. Note that this map is specified by the parameters of the ANN, which are collectively labeled as **w** and indicated with a subscript. The distribution $p_y(\mathbf{y})$ on phase space induced by this map is given by its Jacobian with respect to the input space

$$p_y(\mathbf{y}) = p_y(\mathbf{y_w}(\mathbf{x})) = \left| \frac{\partial \mathbf{y_w}}{\partial \mathbf{x}} \right|^{-1}. \tag{1}$$

The ANN must be trained so that $p_y(\mathbf{y})$ matches the true differential cross section $f(\mathbf{y})$ as closely as possible. As suggested in [1], we can use the Kullbeck-Leibler (KL) divergence $D_{\mathrm{KL}}$

between $p_y(\mathbf{y})$ and $f(\mathbf{y})$

$$D_{\mathrm{KL}}[p_y(\mathbf{y}); f(\mathbf{y})] \equiv \int p_y(\mathbf{y}) \log \frac{p_y(\mathbf{y})}{f(\mathbf{y})} \, d\mathbf{y} \tag{2}$$

to define the loss function to be minimized during training. Since we are working with a discrete set of sampled points $\{\mathbf{x}_i\}$ which are distributed in phase space according to $p_y(\mathbf{y})$, Monte Carlo integration can be performed by replacing the integration with measure $p_y(\mathbf{y}) \, d\mathbf{y}$ by a sum over the sample $\{\mathbf{x}_i\}$ to obtain the loss function

$$L(\mathbf{w}) = \sum_{\{\mathbf{x}_i\}} \log \frac{p_y(\mathbf{y}_{\mathbf{w}}(\mathbf{x}_i))}{f(\mathbf{y}_{\mathbf{w}}(\mathbf{x}_i))}. \tag{3}$$

Note that the loss function should be viewed as a function of the ANN parameters $\mathbf{w}$ with respect to which it will be minimized. For sufficiently large random sample sets $\{\mathbf{x}_i\}$ of fixed size, $L(\mathbf{w})$ will be independent of the sample to a good approximation.

Before describing the structure of the ANN in detail, a few comments are in order. First, the KL divergence has a minimum at zero when the two distributions are identical. However, this assumes that the two distributions have the same normalization. For a given differential cross section, the total cross section is usually not known *a priori*, and thus the loss function will in general have a minimum at some non-zero value. The training procedure, however, depends only on the derivatives of the loss function. Knowledge of the total cross section is therefore not necessary. Second, note that the first equality in Eq. (1) assumes that $y_{\mathbf{w}}(\mathbf{x})$ is a one-to-one map, that is, that each point in target space is reached from only one point in input space. If that is not the case, one would need to sum over the contribution to the induced distribution $p_y(\mathbf{y})$ from each such point. Because the ANN is an arbitrary map, it is possible that it may not be one-to-one. However, if this is the case, it implies that there is a fold somewhere in the mapping. At the location of this fold, the Jacobian vanishes and the induced distribution diverges. We see therefore that if the map is not one-to-one, there must be some places where the ANN is a very poor approximation to the target PDF. Assuming that these places are sampled during the training process, the algorithm will detect the poor fit and attempt to adjust the ANN in order to make it one-to-one. Finally, we note that the map induced by the ANN is not by construction guaranteed to cover all of the target space. However, if at some stage of training it does not cover a part of the target space where the target PDF is non-zero, the loss function would be further reduced by expanding the map to cover more of the target space. We therefore expect that with sufficiently long training, the ANN should learn to cover all regions of the target space that contain significant amounts of the target PDF. For recent alternative approaches that also address these issues, see [4–6].

Each node in a hidden layer of the ANN takes a linear combination of the outputs of the nodes in the previous layer, as determined by the current values of the parameters, and applies a non-linear function known as the activation function. The nodes in the final layer again take a linear combination of the values in the next-to-last layer, but then apply another function, the output function, which is chosen to map onto the set of possible outcomes for the given situation. For our purpose, we need an output function that maps onto the unit interval, since phase space is described as a unit hypercube. Sigmoids are a common choice, but approach the boundary values of 0 and 1 very slowly, making it difficult for the ANN to populate the edges of the hypercube. We try two approaches to deal with this. First, we use a sigmoid as the output function and choose sinh as the activation function. This allows the ANN to easily generate exponentially large internal values, which then allow it to sample the asymptotic regions of the sigmoid. Alternatively, we use the exponential linear unit (ELU)

$$ELU(x) = \begin{cases} x & x > 0 \\ e^x - 1 & x < 0 \end{cases} \tag{4}$$

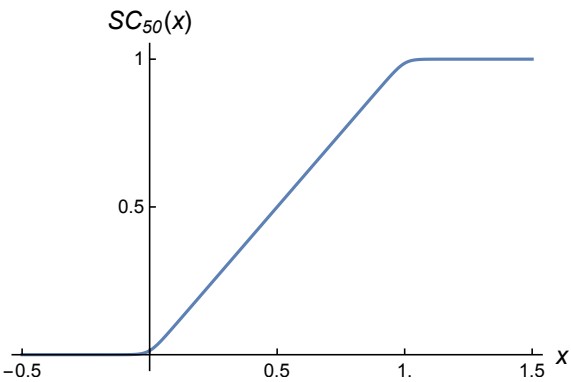

Figure 3: The soft clipping function $SC_p(x)$ given by Eq. (5) with $p = 50$.

as the activation function. The ELU does not generate exponentially large values, so we introduce the *soft clipping function*

$$SC_p(x) = \frac{1}{p} \log\left(\frac{1 + e^{px}}{1 + e^{p(x-1)}}\right) \tag{5}$$

as the output function. This soft clipping function is approximately linear within $x \in (0, 1)$ and asymptotes very quickly outside that range. It is parameterized by $p$ which determines how close to linear the central region is and how sharply the linear region turns to the asymptotic values. A plot of $SC_p(x)$ with $p = 50$ is shown in Fig. 3.

We implement this setup in MXNet 1.1.0 [7] using a Python 2.7 interface. We train with batches of 100 points drawn uniformly over the input space. Each point is mapped onto phase space by the ANN. We then compute the loss function Eq. (3), and take its gradient with respect to the parameters. The gradient is used by the Adam algorithm [8] to update the parameters for the next training batch. Generating points is computationally cheap, so new points are used for each round of training to avoid training onto some random artifact of any one training sample.

Although a loss function like the one described in this section is the appropriate measure with which to train the ANN, for the purposes of assessing the quality of the trained ANN as a MC generator, we compute its efficiency, which is defined as one minus the fraction of points generated by the ANN that must be discarded in order to make the ANN's generated probability distribution match the target distribution. Specifically, for each point in a large sample of size $N$ generated by the ANN, we compute a raw weight defined as

$$w_r(\mathbf{y}) = \frac{f(\mathbf{y})}{p_y(\mathbf{y})}. \tag{6}$$

The probability distribution $p_y$ induced by the ANN may overestimate the target distribution $f$ at some points. In this case we will have $w_r < 1$. In order to correctly match the target distribution, only a fraction of points in the overestimated region should be retained. The appropriate fraction to retain is proportional to $w_r$. However, in other regions, the ANN may underestimate target distribution. The only thing that can be done here is to keep all such points, and scale down the retained fraction in other regions in order to maintain the correct relative shape of the distribution. We must therefore find the maximum raw weight which occurs in the sample, and compute rescaled weights

$$w(\mathbf{y}) = \frac{w_r(\mathbf{y})}{\max(w_r)}. \tag{7}$$

The efficiency $\mathcal{E}$ of the trained NN is then the average value of the rescaled weight over the whole sample

$$\mathcal{E} = \frac{1}{N} \sum_{\{\mathbf{y}_i\}} w(\mathbf{y}_i). \tag{8}$$

It is clear that if the ANN's distribution exactly matches the target then all raw weights will have the same value, which is just equal to the total integral of the target distribution, and thus all rescaled weights are equal to one. Only in this case $\mathcal{E} = 1$.

## 2.2 Phase Space Coordinates

Phase space can be mapped onto a hypercube by the following prescription. For $N$ particles in the final state, we form $N - 2$ subsets: $\{1, \ldots, N-1\}, \ldots, \{1, 2\}$. The invariant mass of a subset $\{1, \ldots, N-k\}$ must lie in the range

$$m_{\{1,\ldots,N-k\}} \in \left( \sum_{i=1}^{N-k} m_i, \sqrt{s} - \sum_{i=N-k+1}^{N} m_i \right), \tag{9}$$

where $m_i$ is the mass of the $i$th final state particle and $s$ is the square of the total center of mass energy of the decay or scattering process. Each of these invariant masses can be linearly rescaled into a variable $q_j \in [0, 1]$, $j \in \{1, \ldots, N-2\}$, so that $q_j = 0\,(1)$ corresponds to the lower (upper) limit of the range (9). Then $2N - 5$ relative angles $\theta_k$ can be chosen between the total momenta of the subsets. These can trivially be rescaled to lie in a unit hypercube. Lastly, an overall rotation can be specified by choosing three Euler angles. In total, this prescription provides $(N-2) + (2N-5) + 3 = 3N - 4$ coordinates as expected.

The VEGAS [2, 3] algorithm tries to adapt a rectangular grid to approximate the desired differential cross section. The phase space coordinates in which the grid is rectangular should be chosen so that any sharp or rapidly varying features in the matrix element are aligned with one of the grid axes. If this is not the case, the VEGAS optimization algorithm is unable to efficiently adapt to it. When a matrix element contains multiple sharp features that are not orthogonal in some coordinate system, it is impossible to make one grid that adapts well to all of them. This is often the case, for example, when a matrix element contains contributions from multiple diagrams that contain different resonant or collinear structures. This is usually handled with multi-channeling [9–11], in which several grids are constructed, each suited to the features in a subset of the diagrams. Each grid is then adapted to approximate the relevant terms in the matrix element, and the relative frequency of sampling from each grid is also adjusted to approximate the full matrix element. (For an alternative approach involving an irregular grid, see [12].)

We note that the ANN has no intrinsic coordinate system or grid. Indeed, the linear transformation followed by the application of a non-linear activation function that occurs between each layer can be viewed as an arbitrary coordinate transformation. The ANN can automatically learn to map the sampling space onto any set of sharp features in the matrix element. We will show examples of this in the following section.

# 3 Examples

## 3.1 Three-body Decay of a Scalar

One of the simplest physical processes we could consider is 3-body decay of a scalar $X$ to three daughter scalars, depicted in Fig. 4(a), with a constant matrix element (which is set to one in what follows). Phase space for this process is 2-dimensional, since the matrix element for the

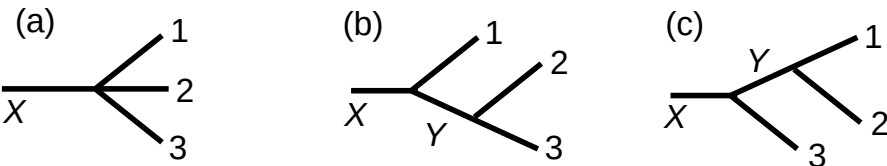

Figure 4: Diagrams of the decay processes used as examples for our ANN.

decay of a scalar particle is symmetric under overall rotations. We choose as coordinates the invariant mass $m_{23}$ of the daughter particles 2 and 3, and the cosine of the angle $\theta$ between the momenta of particles 1 and 2 in the rest frame of $m_{23}$. Both coordinates are rescaled to unit intervals as described above. The differential width to which the ANN is trained is given by

$$d\Gamma = 1 \cdot M^{-1} d\Pi(y_i), \tag{10}$$

where $d\Pi(y_i)$ is the phase space volume element and $y_i$ are the two coordinates. The phase space volume element for three-body decay is independent of the chosen angular coordinate, and is given by

$$d\Pi(m_{23}) = M^{-1} \lambda(M, m_1, m_{23}) \lambda(m_{23}, m_2, m_3) \, dm_{23} \, d\cos\theta, \tag{11}$$

where

$$\lambda(a, b, c) \equiv (2a)^{-1} \sqrt{(a+b+c)(a-b-c)(a-b+c)(a+b-c)}. \tag{12}$$

We assign a mass of $M = 1$ GeV to $X$ and set the daughter scalars' masses to be $(m_1, m_2, m_3) = (0.1, 0.2, 0.3)$ GeV.

We train an ANN with the ELU activation function and the soft clipping output function, and with 3 hidden layers of 128 nodes each. See Appendix A for comparisons of the training performance with other choices of network hyperparameters. The parameters that resulted in the lowest value of the loss function observed during training are saved. The ANN with these parameters is then used to generate a data sample, and this sample is compared to the target distribution in order to compute the unweighting efficiency. After training, the ANN achieves an unweighting efficiency of 75%. For comparison, MadGraph5 [13] achieves an unweighting efficiency of 6% for the same process.

During the early stages of training, the ANN learns the gross features of the target distribution (in this case, for example, that it needs to cover the entire unit square and that it is flat in one direction). In most cases, such general features would be the same even for other choices of the parameters of the physics model (in this case, the masses $M$, $m_i$). As is true for any adaptive algorithm, including VEGAS, it would be faster to re-train an ANN originally trained for one parameter point to another nearby point, relative to starting with a new randomly initialized ANN, given that only small differences between the distributions need to learned. This quality is advantageous in the context of performing parameter space scans that are often necessary in Monte Carlo studies.

## 3.2 Three-body Decay with Intermediate Resonance

To highlight the effect of a non-trivial matrix element, we include an intermediate resonance $Y$ with mass 0.75 GeV in the 3-body decay, either in the form $X \rightarrow 1 + (Y \rightarrow 2 + 3)$ (shown in Fig. 4(b)), or in the form $X \rightarrow 3 + (Y \rightarrow 1 + 2)$ (shown in Fig. 4(c)). As described in section

3.1, for a 2-dimensional phase space, the output coordinates of the ANN were chosen to be the invariant mass $m_{23}$ of final state particles 2 and 3 and a relative angle. This analysis was performed separately for the two decay topologies of Fig. 4(b) and (c). In the first case, the sharp feature in the differential decay width at the location of the $Y$ resonance is aligned with the $m_{23}$ output of the ANN. In the second, the sharp feature appears in the quantity $m_{12}$, which is not used as a coordinate and is not aligned with the axes of the ANN output space.

For this example, we again use an ANN with the ELU activation function and the soft clipping output function, but now with 6 layers of 64 nodes each. With this choice of hyperparameters, the number of weights and biases that specify the network is similar to the configuration used in the last section. However, we note that the non-linearities that enable to the ANN to represent a non-trivial map enter through the activation functions which are present at each layer. Thus, increasing the number of layers is expected to enhance the representational ability of the ANN even if the total number of parameters is not greater. We find that a deeper network provides better performance for adapting to the narrow feature present in this example. The width-to-mass ratio $\Gamma_Y/M_Y$ of $Y$ was varied from $10^{-2}$ to $10^{-4}$ to test the ANN's ability to adapt to features of various sizes. MadGraph5 achieves 6% efficiency for these processes independent of the resonance width. For the aligned resonance of Fig. 4(b), the ANN achieves an efficiency of 71% for $\Gamma_Y/M_Y = 10^{-2}$. For $\Gamma_Y/M_Y = 10^{-3}$ ($10^{-4}$), the ANN efficiency is 53% (29%).

We expect that the level of representational ability needed for the ANN to match a given target distribution well should scale with the inverse of the size of the smallest feature in the target distribution. It is therefore not surprising that the efficiencies are seen to decrease as the width is made narrower, given that we use the same ANN structure for each of these cases. Further improved efficiency could be obtained with more complex ANNs at the expense of increased computational demands.

For the misaligned case of Fig. 4(c), the ANN achieves an efficiency of 32% when $\Gamma_Y/M_Y = 10^{-2}$. The ANN adapts well to narrower widths in this case also, although the required training times are longer than in the aligned case. The lower efficiency in this case is due to the higher complexity of the narrow feature in these coordinates and could be improved further with an ANN with greater representational ability.

In Fig. 5 we show a histogram of the value of $m_{23}$ of events generated with the trained ANN in the aligned example before unweighting compared to the true distribution for the case of a width-to-mass ratio of $10^{-2}$. The raw events match the resonance feature quite well, giving a qualitative indication that large unweighting inefficiencies will not arise. As validation, we also show the histogram after the unweighting procedure has been applied with dashed lines. Unweighting corrects the small residual mismatches in the raw sample to give an accurate fit to the true distribution.

## 3.3 Three-body Decay with Two Resonances

Next, we consider a 3-body decay that can proceed through two diagrams with different resonance structures, that is, two intermediate resonances in different overlapping pairs of final state particles. Specifically, we consider the coherent sum of the processes in Fig. 4(b) and (c) with equal couplings at all vertices. The resonances are given masses of 0.75 and 0.45 GeV, respectively, and widths a factor of 100 smaller than their masses. Given our choice of phase space parametrization, there is no way to align both of these resonance features with the ANN outputs simultaneously. (In this case, since we have only two resonance features and phase space is two dimensional, we could of course choose the two coordinates to correspond to the two resonance features and avoid this problem completely. However, this example is meant to demonstrate a more general point that the ANN approach does not require careful construction of coordinate grids with respect to whatever features are present in the target distribution.)

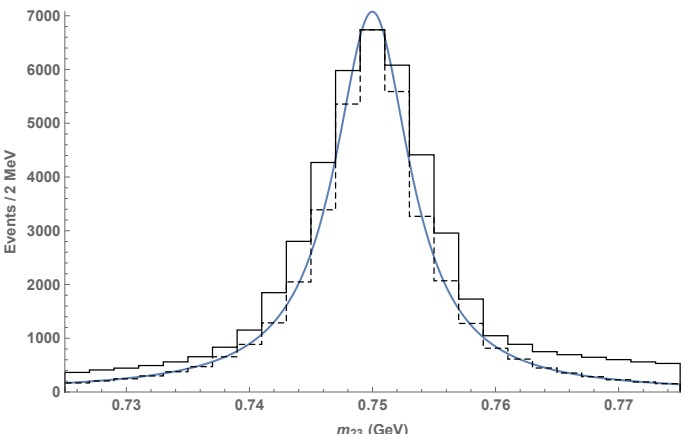

Figure 5: A histogram of the value of $m_{23}$ of $10^5$ events generated with the trained ANN described in section 3.2 without unweighting (solid) and after unweighting (dashed) compared to the true distribution.

With traditional MC generation methods, a situation like this would require the construction of two separate integration channels corresponding to the two diagrams. However, we train our ANN with no modification and find that it performs well, finding both features and distributing events correctly along both. We again use 6 layers of 64 hidden nodes each. To illustrate the advantages of the ANN, we have also trained the VEGAS algorithm on this problem using the publicly available Python implementation [14]. The original VEGAS algorithm is an instance of importance sampling. Recent versions of this code include the ability to augment the original algorithm with a second stage of stratified sampling after the grid has been trained. Because we wish to compare VEGAS with our ANN algorithm, which is also an example of importance sampling, we do not use this additional option in the VEGAS code. We also do not perform any multi-channeling on the integrand.

A sample of events generated by the trained ANN before unweighting can be seen in Fig. 6, where both resonance features are clearly visible. Due to our choice of masses, the partial width of the decay through the aligned resonance is greater than through the misaligned resonance, which explains why the misaligned resonance is less populated. The ANN achieves an unweighting efficiency of 54%, compared to MadGraph5's 6%. Fig. 6 also displays the grid constructed by the VEGAS algorithm. Events generated using this grid would be distributed uniformly in each cell with density inversely proportional to the area. While the algorithm was able to adapt the grid very well to the resonance that is aligned with the coordinate system, it was unable to fit to the misaligned feature. As a consequence, the points generated using the VEGAS grid will give a poor representation of the resonance in $m_{23}$. The unweighting procedure will then require many generated points to be discarded, resulting in poor efficiency. The raw output generated by both the trained ANN and the trained VEGAS grid are shown in Fig. 7. The distribution in $m_{23}$ generated by VEGAS is approximately flat, illustrating its inability to adapt to this feature.

## 3.4  $e^+e^- \rightarrow q\bar{q}g$

The preceding examples contained matrix elements with sharply varying but finite features. Many physically interesting matrix elements also contain singularities, such as the soft and collinear singularities of QCD. As an example, we consider the process of quark pair production from $e^+e^-$ annihilation with an additional final state gluon, and ignoring the contribution of

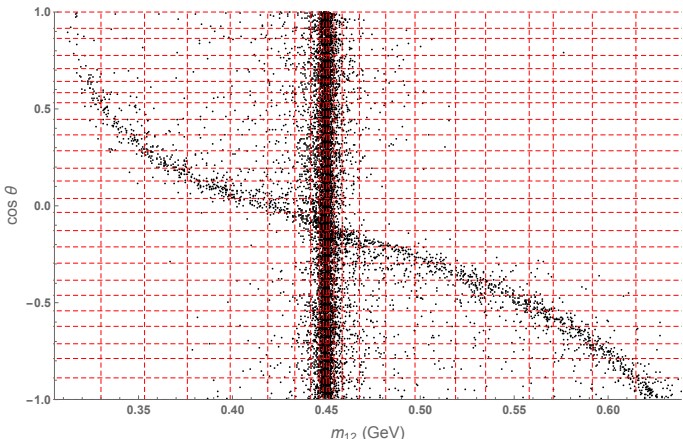

Figure 6: A sample of $10^5$ events generated by the trained ANN described in section 3.3 without unweighting. The matrix element contains two diagrams with different resonant structures. Both are clearly visible in the ANN output. Dashed red lines show the grid generated by VEGAS for the same problem. While the aligned resonance is fit well by the grid, the misaligned resonance cannot be fit.

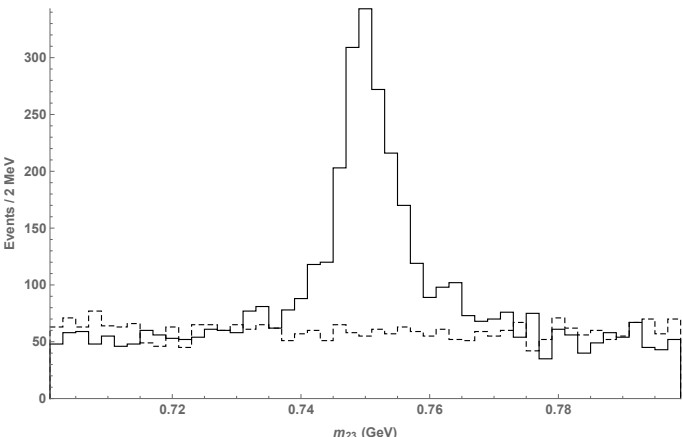

Figure 7: A histogram of the values of $m_{23}$, which corresponds to the resonance which is not aligned with the coordinate system as described in section 3.3, using samples of $10^5$ events generated by the trained ANN (solid) and VEGAS (dashed). No unweighting is performed, so that the figure shows the inherent performance of these two MC sampling methods.

the $Z$ boson. The tree-level differential cross section for this process is proportional to

$$\frac{d\sigma}{dm_{qg}^2 \, dm_{\bar{q}g}^2} \propto \frac{(s - m_{qg}^2)^2 + (s - m_{\bar{q}g}^2)^2}{m_{qg}^2 \, m_{\bar{q}g}^2}, \tag{13}$$

where $s$ is the center of mass energy squared, $m_{qg}$ is the invariant mass of the quark and gluon pair, and $m_{\bar{q}g}$ similarly for the antiquark. The cross section is singular for $m_{qg}, m_{\bar{q}g} \to 0$, and kinematic cuts must be imposed to deal with this singularity. More generally, it is often desirable to impose kinematic cuts, even where singularities are not present, to avoid generating events that are not useful for some reason, *e.g.,* in regions of phase space that lack detector coverage.

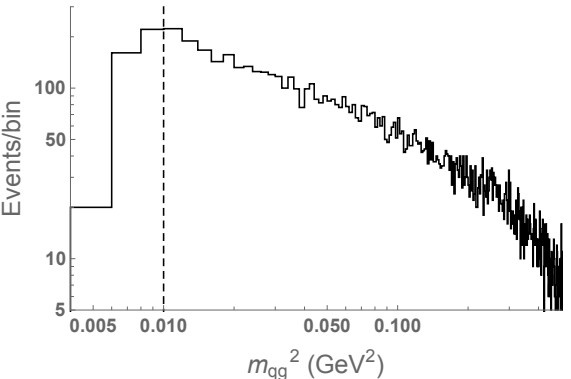

Figure 8: Values of $m_{qg}^2$ of events generated by the ANN trained for the parton-level process $e^+e^- \to q\bar{q}g$ with a cut at $m_{qg}, m_{\bar{q}g} > 0.1$ GeV and $n = 8$ in the cutoff function $\kappa(x)$. The cut location is indicated with a dashed line.

We would like the trained ANN to generate as many events as possible that satisfy the imposed cuts. However, in general it is not possible to train the ANN to completely exclude the cut region. If the value of the target distribution is set exactly to zero in the cut region, the derivative computed during training for any point that falls in the cut region will also be zero. In that case, the trainer will not know how to adjust the parameters of the ANN to bring that point inside the desired region. Likewise, a very sharp change in the target function near the cut threshold will produce very large derivatives there, making it difficult for the trainer to make a final small adjustment to bring a point within the desired region. In other words, the target function needs to always have a relatively moderate slope toward the desired region, so that the trainer can always see in which direction points should be pushed. Because a sharp cutoff cannot be used, the final trained ANN will always generate some undesired points, and these must be manually discarded. Any discarded point decreases the unweighting efficiency, and so one should use a cutoff function that is as sharp as possible while maintaining the ability to train effectively.

If we are cutting on some kinematic quantity $x > x_{\text{cut}} > 0$, a cut function is defined as

$$\kappa(x) = \begin{cases} 1 & x > x_{\text{cut}} \\ (x/x_{\text{cut}})^n & x < x_{\text{cut}} \end{cases} \tag{14}$$

with some $n > 0$. During training, the target differential cross section $d\sigma/dx$ is multiplied by $\kappa(x)$.[1] If the cut is being imposed on a region of phase space containing a singularity in the differential cross section, the value of $n$ must be equal to or greater than the strength of the singularity. During event generation, any event for which $x < x_{\text{cut}}$ is manually discarded.

In the case of the example considered here, we set $s = 1$ GeV$^2$ and take the quarks to be massless. Then $m_{qg}, m_{\bar{q}g} \in (0, 1)$ GeV. Although the cross section takes an especially simple form in terms of the coordinates $(m_{qg}, m_{\bar{q}g})$, we continue to use the general phase space parametrization described in section 2.2, including with the cross section Eq. 13 the appropriate Jacobian factors. We train an ANN with 3 hidden layers of 128 nodes each to this cross section using the same training algorithm described previously. Training takes a similar amount of computational time for this example as well. We find that the ANN is able to adapt to the singular structures in the cross section in these coordinates as well, avoiding the need

---

[1]Note that any multiplicative factor inserted into the target cross section can also be viewed as an additive correction in the loss function Eq. (3), due to the loss function being defined in terms of the log of the induced and target distributions. This cut can therefore be equivalently viewed as a regularizer in the loss function.

to inspect the cross section and choose specific coordinates in advance. We impose the cuts $m_{qg}$, $m_{\bar{q}g} > 0.1$ GeV or $0.01$ GeV as described above, and use various values of $n$ ranging from 4 to 16 in $\kappa(x)$. We find that for all such choices the ANN is able to achieve efficiencies of 65–75%. In this situation, MadGraph5 achieves an efficiency of 4%. A histogram of the values of $m_{qg}^2$ of events generated by the ANN trained with a cut at $m_{qg}$, $m_{\bar{q}g} > 0.1$ GeV and $n = 8$ is shown in Figure 8. The cut location is indicated with a dashed line. The rapid suppression of events below the cut can be seen.

## 4 Conclusions and Outlook

Integrating and sampling differential cross sections and decay widths on multi-dimensional phase spaces is one of the most basic tasks in high-energy physics. Monte Carlo generators provide a numerical tool to tackle this problem. In this paper, we pursued a novel approach to Monte Carlo generation and integration, based on an Artificial Neural Network (ANN) algorithm. The ANN is trained to provide a sample of points in the target phase space distributed according to the known probability distribution function. Distributions that occur in particle physics are often difficult to simulate due to large enhancements of probability in certain special regions of phase space: resonances and soft/collinear singularities. We applied the ANN to examples that contained such features, and showed that the algorithm handles them well.

Most general-purpose tools in use today rely on variations of the VEGAS algorithm for Monte Carlo integration and sampling. Compared to VEGAS, our ANN algorithm offers two advantages. First, the ANN approximates the target distribution by a smooth, rather than piecewise linear, function. This generally allows for a more accurate approximation to be found, yielding higher unweighting efficiency. Second, VEGAS requires a particular choice of coordinates on target space in order to effectively deal with rapidly varying features in the differential cross section. In contrast, the ANN is largely agnostic to the coordinate choice, being able to "learn" the location of the features even if they have a non-trivial shape in a given coordinate system. This adaptability indicates that the ANN algorithm may be preferable in situations with more complicated features, which may occur, for example, in integrating matrix elements beyond leading order in perturbation theory.

In the examples of this paper, the desired probability distributions could be easily computed analytically "by hand." It would be straightforward to interface the ANN algorithm with an automated matrix element evaluation program, such as `MadGraph` [13], to apply it to a broad range of processes of interest. Further, while all examples here were treated at parton-level only, it would be interesting to include parton showering in the same framework.[2] These directions will be pursued in our future work.

## Acknowledgements

The authors are grateful for conversations with Seung Lee, Gabriel Lee, Frank Krauss, and Valentin Hirschi.

**Funding information** We acknowledge the support of the U.S. National Science Foundation through grant PHY-1719877. MDK acknowledges support by the Samsung Science & Technology Foundation under Project Number SSTF-BA1601-07, and a Korea University Grant.

---

[2]For a recent application of ANNs to parton showering, see Ref. [15].

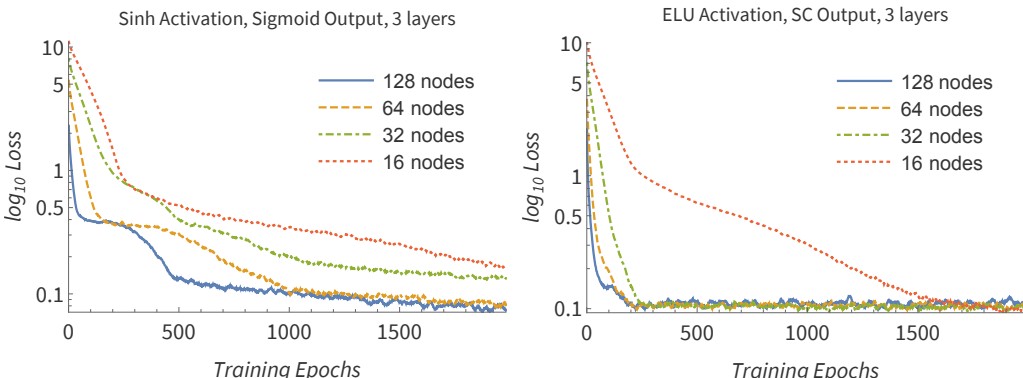

Figure 9: Value of the loss function over the course of training for various choices of the number of hidden nodes per layer in the ANN with a sinh activation function and sigmoid output function (left), or an ELU activation function and SC output function (right). All ANNs have 3 hidden layers.

# A Comparison of network architectures and hyperparameters

In order to study the effects of ANN architecture and hyperparameter choices, we have trained various ANNs and compared their training trends and final efficiencies. For each choice, 10 ANNs were trained on the 3-body decay of section 3.1 with different random seeds. Figure 9 shows the logarithm of the value of loss function obtained at each training epoch for the two choices of activation and output functions introduced in the main text, with various choices for the number of nodes in each hidden layer. Each ANN here has three hidden layers. We plot a running average of the mean of the loss of the 10 ANNs. We observe that while increasing the number of nodes does not decrease the final loss value that can be achieved, it does decrease the number of training epochs needed. (The ANNs with the sinh activation function and 16 or 32 nodes do eventually train to the level of the corresponding ANNs with more nodes, but the plot range has been restricted for ease of comparison with the ELU activation function plot.)

For this simple test case, small ANNs do have the ability to adequately learn the target function. While larger ANNs are therefore not strictly necessary, they do learn faster. In an ANN with the minimal amount of complexity needed to handle a given problem, there would be a discrete minimum of the loss function. Training would start at a random point in the space of ANN parameters and proceed towards this minimum. The space of parameters of an ANN that is larger than necessary can be viewed as a superset of the parameter space of the minimal ANN. The embedding of the minimal ANN in this larger space may not be flat. As such, the distance between a random starting point and a minimum of the loss function in the larger parameter space will usually be shorter than the distance when motion is constrained to the embedded surface of the minimal ANN. This results in faster convergence to low values of the loss function for larger ANNs.

Figure 10 compares the training performance for various choices of the number of hidden layers. Here each hidden layer has 128 nodes. We observe that even for this simple task the performance of the ANN with the sinh activation function degrades rapidly as the number of layers is reduced. However, the ANN with the ELU activation function is robust as the ANN is made shallower. In this example, the ANN with just a single hidden layer actually achieves a slightly lower value of the loss function. We have verified that the resulting unweighting efficiency is indistinguishable from that of the deeper ANNs. On the other hand, deeper ANNs are observed to train in fewer epochs. In the more complicated examples discussed in the main

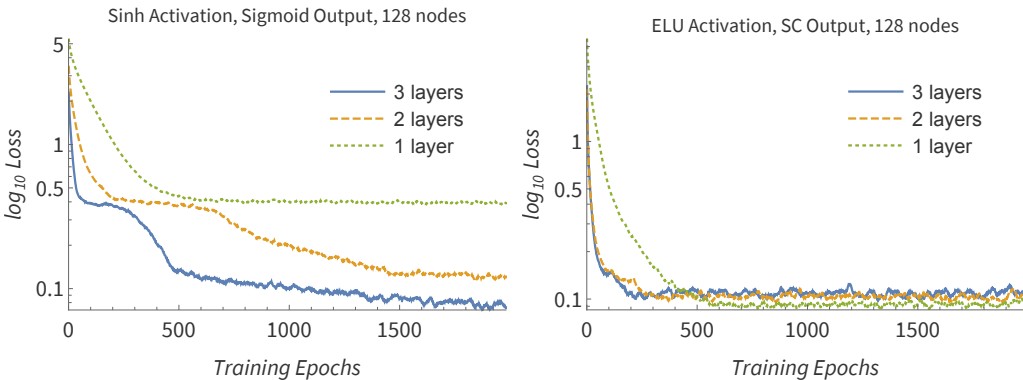

Figure 10: Value of the loss function over the course of training for various choices of the number of hidden layers in the ANN with a sinh activation function and sigmoid output function (left), or an ELU activation function and SC output function (right). All ANNs have 128 hidden nodes per layer.

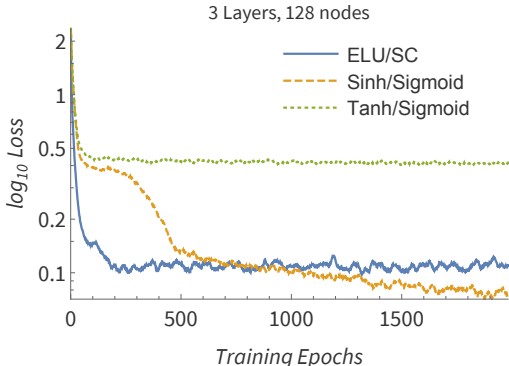

Figure 11: Value of the loss function over the course of training for various choices of the activation and output functions. All ANNs have 3 hidden layers with 128 nodes.

text, one layer is not sufficient, and we indicate in each case the number of layers that is used.

In Figure 11, we compare training performance for the two choices of activation and output functions described in the main text and the choice used in [1], labelled here as "Tanh/Sigmoid." All ANNs have three hidden layers with 128 nodes. We observe that both of the choices introduced in this work significantly outperform that used in [1]. The ELU/SC architecture trains significantly faster than the Sinh/Sigmoid architecture. The Sinh/Sigmoid architecture slowly improves and eventually attains a lower loss value. However, we note that this figure is plotted on a log scale, and the actual difference in the loss values is very small. Moreover, the ELU/SC architecture achieves better unweighting efficiency and has less scatter in unweighting efficiency between different runs.

Based on the preceding observations, for all studies in the main body of the paper, we choose the ELU/SC architecture. The number of hidden layers and nodes used is indicated in each case.

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
