# Peer review of "Neural Network-Based Approach to Phase Space Integration"

_SciPost Physics, doi:SciPost Phys. 9, 053 (2020)_

## Round 2 · Referee Report · Anonymous (Referee 1) · 2020-6-1

Strengths

1) Novel approach 2) Improved efficiency 3) Opens up a new research direction in the field

Weaknesses

1) Implementation of ANN architecture

Report

The paper presents a novel approach for sampling and integrating probability distributions in phase space, suitably parametrized as a unit hypercube. The basic idea is to sample the input space uniformly, and recover the required target distribution $f(y)$ by building a map $y_{\mathbf w}(x)$ with the help of an ANN. This approach is complementary to traditional algorithms like VEGAS and, as the authors have argued and demonstrated with several examples, leads to a higher efficiency. Two additional advantages are: first, the map is smooth as opposed to piecewise linear, which offers a better approximation to the target distribution; and second, the need to use special coordinates aligned with interesting features in the distribution is avoided. In my opinion the paper meets the criteria for novelty and originality, and the method could be usefully applied in support of experimental analyses (after all, it is the experimentalists who are the end users of such large MC production jobs). In addition, the paper has already had an impact by inspiring several follow-up studies (including a paper published in SciPost), which further expanded and refined the method. Therefore I think the paper can be published.

Requested changes

I have only a few comments and recommendations which the authors should consider before the final version.

  1. Ideally, it would be great if the authors can implement the method using a reversible neural network architecture - this is needed in order to properly compute the Jacobian (1), as discussed in subsequent works 2001.05478 and 2001.05486. At the same time, considering that the purpose of the paper is only to demonstrate the viability of this approach (which the authors have) and not to provide a deployment-ready final product, such an exercise is not crucial to the utility of the paper to the community. A disclaimer (in the introduction and conclusions) that reversible networks should be used (or a rough argument why the used architecture is unlikely to map two inputs to the same value) will be sufficient to warrant publication. Along those lines, the authors should perhaps also add a comment on the surjectivity issue brought up by 2001.05478 (Sec. 2.3).

  2. The comparison in Fig. 5 is only qualitative - the two distributions are not supposed to match exactly, although the text might leave this impression: ``the raw events match the resonance feature very well''. To avoid confusion, perhaps add the words "before unweighting" to the figure caption as well, and expand the discussion in the text to clarify the exact purpose of the comparison shown in the plot. As a validation check, is it possible to add a second panel showing distributions after unweighting, e.g., comparing VEGAS, ANN and the true distribution?

  3. The authors have focused on the efficiency as their performance benchmark. Yet other benchmarks are also worth exploring - for example, the smoothness of the approximation was mentioned a few times as an advantage over VEGAS, so it would be nice to see an example quantifying the improvement (if any) in terms of the precision (on the calculated integral or on the shape of the generated distribution) which can be achieved with the proposed method over VEGAS or similar piecewise linear approximations.

  4. There is a typo on page 5 line 7 from the top: "and indicateD with a subscript". Also in several places in the text there is a reference to "the A". This could be the journal style, but perhaps it can be clarified that what is meant is "Appendix A" and not just "The A".

  • validity: high
  • significance: high
  • originality: top
  • clarity: high
  • formatting: excellent
  • grammar: perfect

Author:  Matthew Klimek  on 2020-08-18  [id 930]

(in reply to Report 1 on 2020-06-01)

Thank you for the comments.

  1. These comments are valid, and we have added a discussion of them in a new paragraph on page 5.

  2. We have made the suggested clarification in the caption of Figure 5 and in the last sentence of Section 3.2, and added a histogram of the unweighted events to Figure 5, with accompanying mention in the text.

  3. In response to a comment in the second referee's report, we have added a visualization of the piecewise-constant approximation generated by VEGAS to Figure 6, and a new visualization of the raw output of both ANN and VEGAS for the misaligned resonance described in Section 3.3 in Figure 7. This provides a graphical indication of the smoothness advantage of the ANN. We hope that this will help address this point. Quantitatively, any benchmark that quantifies the accuracy of the approximation will be a function of the ratio of the target PDF and the PDF of the (ANN or VEGAS) approximator. If the approximation becomes perfect, this ratio will be 1 everywhere (up to normalization), the efficiency will be 100%, and any other conceivable benchmark of this kind will also be maximized. We therefore keep efficiency as our benchmark metric, but hope the additions to the figures will provide additional qualitative insight. Regarding the precision of the integral, this can always be improved by evaluating more points, even if a poor approximation to the target function is used. For a fixed number of points, a better approximation will give a more precise estimate, but this is again quantitatively related to the efficiency.

  4. Thank you for pointing out these typos. They have been corrected.

---

## Round 2 · Referee Report · Anonymous (Referee 2) · 2020-6-4

Strengths

This paper gives a new technique to perform phase space integration using artificial neural networks.

1) This is an indeed an interesting subject, and more efficiently sampling phase space in monte carlo event simulations to obtain unweighted events would be useful to the community. These types of tools are used by a substantial number of particle phenomenoligists, and their improvement would make an impact.

2) Artificial neural networks are a relatively new area of study in particle physics, and applying them to monte carlo simulations is promising. This paper is new work, showing potentially interesting application of these new methods. Artificial neural networks may improve existing new tools, and contributions such as this study is worthwhile.

3) The authors show that artificial neural networks can efficiently generate unweighted events for integrands with resonance or singularities, even without variable transformations to different coordinate systems to more efficiently sample sharp peaks.

4) The method presented here is a continuous mapping of phase space, unlike the usual discrete grid of phase space used by VEGAS. These types of methods could promise to create a much more efficient sampling of phase space unlike the usual discrete sampling.

5) The authors explore many different 3 body final states that present different issues with event generation: resonance and collinear/soft signularies.

Weaknesses

The method of this paper appears solid. My major concerns are the comparisons between the artificial neural networks and more traditional methods, and the presentation of results.

1) The authors consistently compare their unweighting efficiency to MadGraph's (MG's) unweighting efficiency. It is not clear this is an apples to apples comparison.

1a) First, it's not clear if the unweighting efficiencies of the artificial neural network (ANN) and MG are before or after training. For a fair comparison, they should either be both before or after training the integration grid of ANN.

1b) Second, the authors specifically design their ANN for a 3-body final state while MG is for generic N-body final states. As the authors do in their paper, a 3-body phase space can be written as a 2-dimensional integration. Hence, all sampled points are physically allowed, which can increase efficiency. MG is designed for a general final state and not specifically a 3-body final state, which may add additional inefficiencies beyond the unweighting procedure.

2) The presentation of results could be be better. There are several plots showing results. However, there is only one comparing the ANN output with the true distribution, and no plot comparing the ANN with more traditional methods.

Report

The authors present a promising new method to perform monte carlo simulations in particle physics by using artificial neural networks (ANNs). The proposal is to ues an ANN to efficiently sample phase space and generate unweighted events. In traditional monte carlo methods, it can be difficult to efficiently sample regions where an integrand is sharply peaked. This happens many times in particle physics, for example when there are narrow resonances or collinear and soft singularities. A major hurdle is that monte carlo integrators such as VEGAS use a grid to sample the phase space. As the authors point, an advantage of an ANN is that it doesn't use a grid but samples phase space continuously.

This paper is well written and timely. The authors apply their method to many different benchmark scenarios in particle physics involving: both narrow resonances and collinear/soft singularities. The analysis, the ANN, and the motivations are well explained. However, the comparison with more traditional VEGAS monte carlo methods is lacking. This is detailed under the "Weaknesses" portion of this report. That being said, it is straightforward to have an apples to apples comparison between the ANN and VEGAS methods. Under "Requested changes" there are several suggestions to help with these corrections, as well as suggestions of plots to illustrate the comparisons. Once these changes are implemented, the paper will be ready to publish.

Requested changes

As mentioned above, it does not seem the authors are performing an apples to apples comparison between their proposed methods and traditional VEGAS methods. Here are several requests for clarification, new plots, and different comparisons.

1) On the bottom of pg. 8 the authors say "We therefore expect that, with an ANN trained for one parameter point, it would be relatively fast to re-train that ANN for a nearby parameter point compared to starting with a randomly initialized ANN, ..." However, this is also true of traditional VEGAS methods, where you can save the grid as a "statefile." These statefiles can be also be used for different parameter points as along as the phase space is the same. The authors should make clear this is not just a feature of the ANN.

2) The authors should make it clear if the quoted ANN unweighting efficiency is before or after the training of the neural network.

3) In light of points (1,2), the comparison I think should be made is between the unweighting efficiencies of the ANN and VEGAS after the training of both. Three body phase spaces are simple enough that it is straightforward to use VEGAS for the phase space integration. Indeed, the authors already list which variables they are using and how to map them onto a unit square, and this can be implemented into VEGAS as well as their ANN. By training VEGAS, I mean precisely point (1). An initialization run can be used to create a VEGAS grid. Then calculating the unweighting effeciency with a VEGAS run with the initialized grid as the input. This would be an apples to apples comparison of two dedicated 3-body monte carlos, and would be a more fair comparison that using MadGraph.

4) There should be plots comparing the distributions obtained between traditional VEGAS methods and the ANN. For example, Fig. 5 could show both the VEGAS and ANN histograms. This would show explicitly how well the two methods sample the phase space. This comparison should be made for all plots in Fig.s 5,6,7.

5) In Fig. 6, as the authors say, it is clear that the ANN is finding both the aligned and unaligned resonances. It would be very helpful to explicitly show the 1-D unaligned invariant mass distribution for both the ANN and VEGAS. This would explicitly show how well the ANN and VEGAS are reconstructing that invariant mass.

6) This one is purely optional. For Fig. 7, the authors point out that enforcing a cut by setting the integrand to zero can make the training of the ANN problematic. Hence, they introduce the function in Eq. (14) to make the cuts continuous. It would be nice to see the comparison between the current Fig. 7 and strictly setting an integrand to zero to explicitly illustrate this point.

  • validity: high
  • significance: good
  • originality: high
  • clarity: high
  • formatting: excellent
  • grammar: excellent

Author:  Matthew Klimek  on 2020-08-18  [id 929]

(in reply to Report 2 on 2020-06-04)

Thank you for the comments.

  1. It is of course correct that this is not an exclusive feature of the ANN, and we did not intend to imply that. We have modified the last paragraph of 3.1 as suggested.

  2. Our original text specifies that the efficiency is computed after training: "After training, the ANN achieves an unweighting efficiency of 75%."

  3. We understand the referee's interest in comparing our method directly to VEGAS, rather than to a general-purpose tool such as MadGraph. However, we wish to point out that the efficiency achieved by VEGAS will be a function of the number of bins that are used, which is a free parameter. As we discuss in the text, one can view our method as the continuum limit of VEGAS, and so we could in fact achieve similar efficiency within VEGAS by using a very large number of bins. Because there is no natural choice for the number of bins, there is no invariant notion of comparison between our method and VEGAS. Moreover, increasing the number of VEGAS bins will also increase the amount of CPU time needed to adapt the grid, an accounting of which should be factored into any fair comparison. So the precise comparison will always depend on the parameters and details of the training procedure. Because of this we did not include a comparison between our method and VEGAS with some ultimately arbitrary parameters. Instead, we chose to compare with a standard widely-used tool that everyone in the community will be familiar with. We hope the referee will find this viewpoint acceptable.

  4. In the case of Figure 5 (showing the single aligned resonance), as stated above, one could make VEGAS arbitrarily good by increasing the number of bins, and so for the same reasons we chose not to make that comparison. However, in the case of the example with two resonant structures (shown in Figure 6), there is a fundamental difference. As we emphasize in the text, VEGAS needs strong features to be aligned with its coordinate system. Increasing the number of bins will do nothing to help with this. We therefore agree that for this case, the comparison is interesting and has definite meaning regardless of the choice of VEGAS parameters. We have added grid lines to Figure 6 showing the VEGAS grid after training.

  5. As above, we agree that this comparison is interesting and we now show this in a new Figure 7.

  6. It is actually simply not possible to train in the case when the integrand is set to zero in some region of measure >0. This is because the derivative of the loss function with respect to the weights will be zero for points in that region. A zero derivative will then cause the trainer to try to make an infinite change in the weights. So we cannot even make such a plot for comparison. We apologize if this was not clear from the text.

---

## Round 3 · Referee Report · Anonymous (Referee 3) · 2020-9-2

Strengths

1) Novel method
2) Improves the efficiency for phase space integral, has the potential to break one of the bottleneck for particle generators
3) A thorough study of various 3-body structures

Weaknesses

1) This study mainly shows improvements compared to non-NN bases methods, which is generically anticipated. It would be of much more importance and impact if the author explores which NN structure would be (quasi-)optimal for this important physics question of physics space integration for particle physics.
2) The author explores 3-body final states. It would be helpful to common on the expected scaling behavior once applied to a higher number of final state particles.

Report

The authors performed a novel study using ANN to perform phase space integral, showing promising features when compared with more traditional methods. The study is robust and has great potential for MC generation applications.
The manuscript is of high quality and provides insights on an important topic. The authors have carefully addressed all the issues and suggestions raised in the previous reports. I recommend direct publication at SciPost.

---

## Round 3 · Referee Report · Anonymous (Referee 4) · 2020-9-5

Strengths

1) Novel approach
2) Improved efficiency
3) Opens up a new research direction in the field

Weaknesses

1) Implementation of ANN architecture

Report

The authors have addressed the points in my previous report and the paper is now ready for publication in SciPost.

Requested changes

None.

---

## Round 3 · Author Response

We thank the referees for their helpful comments. Based on their feedback, and the editor's recommendation, we have incorporated a number of minor improvements to the draft, while the results and conclusions remain unchanged. Those improvements are listed here, and more detailed responses to the comments will be submitted below each referee report.

---

## Round 3 · List of Changes

- A typo above Eq. 1 was fixed.
- A new paragraph has been added starting below Eq. 3 discussing issues related to the bijectivity on the ANN.
- The discussion in the last paragraph of Sec. 3.1 has been modified to clarify that the ability to retrain quickly to nearby points in parameter space is not exclusive to the ANN algorithm.
- Fig. 5 now also shows the ANN output after unweighting, as validation that the unweighting procedure corrects any residual mismodeling. The last paragraph of Sec. 3.2 has been lengthened to describe the figure clearly.
- Grid lines have been added to Fig. 6, showing the result of applying VEGAS to this problem. The corresponding description of the plot in the caption and main text have been updated.
- There is a new Fig. 7 showing the raw output of the ANN and of VEGAS for the resonance that is not aligned with the phase space coordinate system. This is also described in the main text.
- The word "Appendix" has been added before references to the appendix.

---

## Editorial Decision

published